# Training and Adaptation of Beef Calves to Precision Supplementation Technology for Individual Supplementation in Grazing Systems

**DOI:** 10.3390/ani13182872

**Published:** 2023-09-09

**Authors:** Joshua L. Jacobs, Matt J. Hersom, John G. Andrae, Susan K. Duckett

**Affiliations:** 1Department of Animal and Veterinary Sciences, Clemson University, Clemson, SC 29631, USA; jjacobs9@g.clemson.edu (J.L.J.); mhersom@clemson.edu (M.J.H.); 2Department of Plant and Environmental Sciences, Clemson University, Clemson, SC 29631, USA; jandrae@clemson.edu

**Keywords:** precision supplementation, technology adaptation, beef cattle

## Abstract

**Simple Summary:**

Understanding the relationship between supplementation and animal production requires a systems approach to understand all variables affecting this relationship. Many publications regarding the utilization of consumed supplements have been published, as have studies focused on cattle production response. Supplementation costs strongly influence farm profitability, yet many supplementation strategies aim to supply nutrients to the average of the group, though inter-animal variation within groups can differ vastly meaning both over- and under-supplementation is occurring, potentially decreasing production. Measuring real-time cattle intake is historically labor intensive and typically focuses on total intake instead of individual supplement intake. New precision feeding technologies such as the C-Lock SuperSmart Feeder (SSF, C-Lock Inc., Rapid City, SD, USA) allow for real-time supplement intake data collection in field settings but require adaptation periods to facilitate the usage of these technologies by cattle. New technologies, such as the SSF, offer insight into behavioral information relating to cattle adaptation rates, feeding patterns, and the relationship between supplementation and animal production. The objective of this research was to assess the training and adoption rates of three different groups of cattle (suckling calves, weaned steers, replacement heifers) to the SSF.

**Abstract:**

Supplementation of beef cattle can be used to meet both nutrient requirements and production goals; however, supplementation costs influence farm profitability. Common supplementation delivery strategies are generally designed to provide nutrients to the mean of the group instead of an individual. Precision individual supplementation technologies, such as the Super SmartFeed (SSF, C-Lock Inc., Rapid City, SD, USA), are available but are generally cost prohibitive to producers. These systems require adaptation or training periods for cattle to utilize this technology. The objective of this research was to assess the training and adoption rates of three different groups of cattle (suckling calves, weaned steers, replacement heifers) to the SSF. Successful adaptation was determined if an individual’s supplement intake was above the group average of total allotted feed consumed throughout the training period. Suckling calves (n = 31) underwent a 12 d training period on pasture; 45% of suckling calves adapted to the SSF and average daily intake differed (*p* < 0.0001) by day of training. Weaned steers (n = 79) were trained in drylot for 13 d. Of the weaned steers, 62% were trained to the SSF, and average daily intake differed (*p* < 0.0001) by day of training. Replacement heifers (n = 63) grazed tall fescue pastures and had access to SSF for 22 d of training. The success rate of replacement heifers was 73%. For replacement heifers, the daily intake did not differ (*p* < 0.0001) by day of training. Results indicate production stage may influence cattle adaptation to precision technologies.

## 1. Introduction

The supplementation of beef cattle can be used to reach nutrient requirements not met by forage alone or to increase animal production to achieve desired production goals [1]. Supplementation strategies influence farm profitability, and due to the recent cost increase in many commonly used supplement sources, precision supplementation is of major importance [2]. Precision livestock feeding, specifically through the use of sensor-based technologies, can be utilized to match nutrient supply to the individual’s requirement in real time [3]. By increasing the precision of beef cattle supplementation, nutrient requirements can be met more precisely and excess nutrient use can be avoided [4]. Farm profitability improves when overall supplementation costs can be minimized, and animal performance can be optimized by avoiding excess nutrient consumption. Many supplementation strategies are designed to provide nutrients to the mean of the group rather than to an individual animal [5]. Measuring individual intake in production settings is difficult at best, but precision supplementation technologies allow this information to be gathered primarily in research settings [6,7]. The information gathered in research settings may then be disseminated to producers for application in production settings.

Understanding the relationship between feeding behavior and nutrient utilization is crucial to improving precision supplementation and total feeding strategies. A variety of precision feeding technologies have been developed to help understand feeding behaviors and collect real-time intake measurements [8]. Many of the precision feeding technologies are better utilized for total intake data collection and are relatively stationary, requiring a permanent connection to electricity or consistent access to networks for data collection. Older technologies such as Calan Gates (American Calan Inc., Northwood, NH, USA) require daily manual refilling and data collection. More recent technologies such as GrowSafe (GrowSafe Systems Ltd., Airdrie, AB, Canada) or SmartFeed (C-Lock Inc., Rapid City, SD, USA) offer automated intake data collection, but still require frequent refilling. In contrast, the Super Smart Feeder (SSF, C-Lock Inc., Rapid City, SD, USA), is a more mobile precision feeding technology that focuses more on supplementation; allows controlled feed allotment, in-field data collection, and in-field supplement delivery; and requires less manual refilling.

Cattle often require adaptation or training periods to acclimate and utilize precision feeding technologies, much like training to utilize conventional feed bunks [9]. Despite being crucial for accurate data collection, little information is available defining specific training or adaptation methods and the success of these methods. Both supplement intake and foraging behavior have been shown to be influenced by the age of cattle, leading to further questions regarding feeding behavior when utilizing precision feeding technologies [10]. The objective of this research was to assess the training and adaptation of beef cattle in three different production stages (suckling calves, weaned steers, replacement heifers) to the SSF.

## 2. Material and Methods

### 2.1. Experimental Site

Three experiments were conducted at the Piedmont Research and Extension Center, Pendleton, SC, USA, to evaluate the adaptation of beef cattle in various production stages to precision feeding technologies. All animal experimental procedures were reviewed and approved by the Clemson University Institutional Animal Care and Use Committee (AUP2021-0138, AUP2021-0044, and AUP2020-0041).

### 2.2. Super SmartFeeder (SSF)

The SSF is a solar-powered, automated, mobile precision feeder consisting of a four-chambered feed bin capable of dispensing up to four supplement types into four separate, individual feeding stalls (Figure 1). The presence of radio frequency identification tags assigned to each individual animal triggers feed dispersal as described by [11]. The SSF utilizes a cloud-based interface to collect and store data to allow users to determine individual animal supplement intake, the number of animal visits to the feeder, and the timing of animal visits. The SSF technology also allows researchers to limit individual animal supplement intake by assigning individual supplement allotments as well as exclude individuals from accessing the SSF.

Proper set-up of the SSF unit is imperative to ensure accuracy. The SSF was calibrated prior to use per manufacturer recommendations. The SSF feeder was set to dispense small amounts of supplement (<0.1 kg) for each drop when the animal accessed the feeder. The weight of the supplement is recorded for each drop while the animals continue to access the SSF. The animals can only consume a maximum intake for each day as defined in the system software by the experimenter. The SSF solar panel was oriented to face south to ensure adequate charging. Additionally, each time the SSF unit was refilled or moved following initial calibration, the unit was leveled and feed drops of each feeding stall were manually calibrated per manufacturer recommendations on an independent scale to ensure accuracy within the SSF system.

The success of training for all experiments was defined as the total individual intake greater than or equal to the average percentage intake of total supplement offered to the respective group. Meeting this parameter helps identify individuals within their respective groups that are more likely to consistently utilize the SSF. Failure to train was defined as the total individual intake less than the average percentage intake of the total supplement offered to the respective group.

#### 2.2.1. Experiment 1

Suckling calves (n = 31, age = 86 ± 13 d, BW = 133.5 ± 15.6 kg) were allowed access to all four SSF stalls for a 12 d training period in a 4.1-ha tall fescue pasture. The SSF stall gates remained down for the duration of training to prevent dams from accessing supplement and ensure individual access to the SSF. Suckling calves received 0.45 kg of a commodity mixed feed (J & D-Lancaster Inc., Lancaster, SC, USA) from d 0 through d 3, and were then transitioned to 0.45 kg of steam-rolled corn (Godfrey’s Feed, Madison, GA; 87.4% DM, 7.3% CP, 83.8% TDN) on d 4 where they remained throughout the remainder of the training period. The commodity mixed feed was offered as a palatable starting feed to encourage suckling calf consumption from the SSF, since the suckling calves used for this experiment had no previous supplement exposure.

#### 2.2.2. Experiment 2

Angus-crossed steers (n = 79, age = 188 ± 23 d, BW = 227.5 ± 35.8 kg) were weaned and placed into a 1.0-ha drylot with access to all four SSF stalls for a 13 d training period. Steers were allotted 2.27 kg of a commodity mixed feed (J & D-Lancaster Inc., Lancaster, SC, USA; 89.3% DM, 18.0% CP, 75.1% TDN) from the SSF daily. Additionally, steers were allotted 2.27 kg of the commodity mixed feed in concrete feed bunks at 1400 h daily throughout the training period and had ad libitum access to bermudagrass hay (86% DM, 15.6% CP, 58.9% TDN) and water. From d 0 through d 3, SSF stall gates were raised, and on d 4 gates were lowered to ensure individual access to the SSF. The SSF stall gates remained down throughout the remainder of the training period. Steers used for this experiment had no exposure to supplementation prior to the start of training.

#### 2.2.3. Experiment 3

Replacement heifers (n = 63, age = 255 ± 20 d, BW = 267 ± 31.7 kg) were allowed access to the SSF for a 22 d training period in a 4.1-ha tall fescue pasture. Heifers were allotted 3.64 kg of a commodity mixed feed daily (J & D-Lancaster Inc., Lancaster, SC, USA; 90.0% DM, 16.3% CP, 68.7% TDN). Replacement heifers had no other access to supplements. From d 0 through d 5, SSF stall gates were raised, and on d 6 gates were lowered to ensure individual access to the SSF. The SSF stall gates remained down throughout the remainder of the training period. Heifers used in this experiment had previous supplementation exposure throughout the weaning period, prior to the initiation of training to the SSF.

### 2.3. Statistical Analysis

A chi-squared test was performed using the FREQ procedure of SAS Version 9.4 (SAS Inst. Inc., Cary, NC, USA) to determine differences in the frequency of training success and failure. Success was defined as a total individual intake greater than or equal to the average percentage intake of total supplement offered to the respective group. Non-feeders were included in all statistical analyses. The percent of maximum allotted supplement cattle consumed was analyzed using the GLM procedure of SAS with training outcome in the model. Individual daily intake was analyzed using the GLM procedure of SAS with day of training in the model. Least square means were generated and separated using the PDIFF option of SAS. Significance was determined at (*p* < 0.05). Individual animals were the experimental unit for all studies.

## 3. Results and Discussion

No statistical difference (*p* = 0.590) was observed for the frequency of training success or failure in suckling calves. An adaptation rate of 45% was observed for suckling calves. Their individual average percentage intake of total supplement offered relative to the group average of 41.84% is depicted in Figure 2A. An average intake of 66.04% ± 3.41% and 21.92% ± 3.10% of total offered supplement was observed for adapted and non-adapted calves, respectively (*p* < 0.001). Figure 2B depicts individual calf intake by day of training relative to the average daily intake of the group. Suckling calves had an overall average daily intake of 0.30 kg. Average daily intake differed by day (*p* < 0.001) for the sucking calf training period (Figure 2B). Average daily intake was lowest on d 1 and increased daily until d 3. On d 4, average daily intake decreased sharply, likely due to the change in feed offered by the SSF. However, consumption increased on d 5, peaking on d 6 and did not differ from the remainder of the training period. The literature discussing suckling calf adaptation or utilization of in-field precision supplementation technologies, such as the SSF, was unavailable. Collecting data on cattle in this production stage is difficult due to calf reliance on the dam. Suckling calf movement throughout paddocks is likely dictated by the dam as well. To help facilitate suckling calf interaction with the SSF, the cattle were placed in smaller 4.1-ha paddocks to increase the proximity of calves and dams to the SSF.

A greater (*p* = 0.033) number of successful training outcomes was observed compared to the failure-to-train outcomes for weaned steers in Experiment 2. Weaned steers exhibited an adaptation rate of 62%. Their individual average percentage intake of the total supplement offered relative to the group average of 49.78% is depicted in Figure 3A. An average intake of 70.18% ± 1.99% and 16.46% ± 2.54% of total offered supplement was observed for adapted and non-adapted calves, respectively (*p* < 0.001). Figure 3B depicts individual steer intake by day of training relative to the average daily intake of the group. Weaned steers had an overall average daily intake of 1.59 kg from the SSF. Average daily intake differed by day (*p* < 0.001) for steers and is illustrated in Figure 3B. Average daily intake was lowest on d 1. Steers reached their maximum average daily intake on d 10; however, there were no statistical differences from d 9 through d 11. A decrease in average intake occurred on d 12 of training but rebounded on d 13. The decrease was likely due to the SSF being refilled.

The frequency of successful training outcomes observed for replacement heifers was greater (*p* < 0.001) than the frequency of failure to training outcomes. Replacement heifers exhibited an adaptation rate of 73%. Their individual average percentage intake of total supplement offered relative to the group average of 70.64% is depicted in Figure 4A. An average intake of 90.02% ± 3.37% and 18.95% ± 2.05% of total offered supplement was observed for adapted and non-adapted heifers, respectively (*p* < 0.001). Figure 4B depicts individual heifer intake by day of training relative to the average daily intake of the group. The average daily intake of the training period was 3.23 kg. Replacement heifer average daily intake did not differ statistically by day (*p* = 0.075) throughout the training period. Numerical differences were observed and are depicted in Figure 4B. Replacement heifers had the lowest average daily intake on d 1 of training, and average daily intake increased numerically on d 2 and peaked on d 10. A sharp numerical decrease in average daily intake was observed on d 12 and decreased until d 13. This is likely due to the SSF being refilled. However, average daily intake appeared to increase to levels similar to what was observed prior to the numerical decrease on d 12.

Although several studies regarding the utilization of precision feeding technologies have been published, little literature is available regarding training success and failure rates for beef cattle of any production stage with SSF units. Several reports describe training procedures. A study by Husz et al. [11] utilizing preconditioned beef steers (n = 418, 8–10 months of age) reported a 7 d training and adaptation period for steers in each year of a two-year study. Success was defined as a steer consuming 0.5 kg of supplement for a 3 d period. Unlike the current studies, steers were exposed to the SSF in groups of 40 to 45 animals to limit competition. Husz et al. [11] immediately removed trained steers from the group to further limit competition at the feeder. Steers not achieving the criteria within 7 d were considered ‘non-feeders’. Throughout the study, Husz et al. [11] reported 31% of steers did not voluntarily use the SSF following the training period but did not report non-feeder numbers through the training period. Limiting initial competition of precision supplementation technologies like the SSF may improve the success of adaptation periods by allowing more opportunities to feed.

Similar to the weaned steer study, a study by Valliere et al. [12] reported a 24 d acclimation period for post-weaning lambs (n = 179) on pasture. Lambs were offered 0.23 kg of supplement daily from an SSF in a pasture setting. From d 0 through d 10, lambs had access to 0.45 kg of supplement in standard troughs, decreasing to 0.15 kg of supplement. From d 0 through d 5, Valliere et al. [12] reported 30 and 80 percent of lambs visited the SSF. However, lamb SSF visits varied between 64 and 78% after d 6. From d 6 to d 24, 72% of lambs visited the SSF daily. The success rate of post-weaning lambs reported by Valliere et al. [12] is greater than that observed in weaned steers in Experiment 2. Both the current studies and the study reported by Valliere et al. [12] suggest longer acclimation periods may prove helpful in improving the utilization of precision feeding technologies.

A study by Williams et al. [13] reported a 35 d training period to allow steers (n = 40, BW = 243 ± 23 kg) to acclimate to a similar automated feeder, SmartFeed (C-Lock Inc., Rapid City, SD), by offering 0.91 kg of supplement three times per week to encourage use of the feeding system. Williams et al. [13] reported a 12.5% non-feeder rate throughout the following study (n = 16) but did not report training results. A similar study by McCarthy et al. [14] reported a 14 d training period to a SmartFeed system utilizing yearling heifers (n = 126, BW = 400.4 ± 6.2 kg), and stated that non-training heifers were then utilized as control animals in the study following training. Another study by Stewart et al. [15] reported a 10 d acclimation period to allow mature ewes (n = 78) to acclimate the SSF. There was no mention of adaptation success or failure rates in these studies. These studies reported using older animals; however, the absence of training success and failure in the reports makes it difficult to compare the results of the current study using replacement heifers.

The age or production stage of cattle appears to influence adaptation time. In the current studies, replacement heifers that had been previously exposed to supplementation responded with a rapid increase in average daily supplement intake, reaching daily intakes greater than 90% of allotted supplement as early as d 2 of exposure to the SSF. Suckling calves and weaned steers naive to supplementation had slower increases in average daily supplement intake. Cattle that have never been exposed to supplementation of any kind not only have to adapt to the SSF but also the supplement itself, which may explain the increase in average daily supplement intake being less than that observed in the replacement heifer study.

## 4. Conclusions

Prior to data collection from precision feeding technologies such as the SSF, cattle must be allowed to train and adapt to utilizing such technologies to ensure the data collected are accurate. Adaptation rates of cattle appear to differ based on the production stage. Cattle previously exposed to supplementation (replacement heifers) appear to adapt quicker than cattle naive to supplementation (suckling calves and weaned steers). Suckling and weaned calves naive to supplementation may require training periods of 7 d or more, while older cattle with previous supplement exposure may be able to train in less than 7 d. However, allowing additional time during training may help identify animals that will utilize the SSF more consistently throughout the subsequent study. Introducing cattle to the SSF with the stall gates raised may hasten adaptation. However, in situations like the suckling calf study, this was not an option; therefore, longer training periods may be required. Animal proximity to SSF units may also help increase the success of adaptation. If possible, training or adaptation periods should be performed in smaller paddocks or drylots to increase animal interactions with the SSF. Placing SSF units on animal travel routes, i.e., between loafing, grazing, and watering areas, may help increase cattle interactions with the SSF. Further advancement and use of precision feeding technologies will enhance nutritional management and encourage more efficient nutrient utilization by ruminant animals. These advances will help improve supplementation recommendations, minimize wastage of expensive feedstuffs, enhance individual animal performance, and increase research opportunities utilizing precision supplementation technologies. More research regarding animal behavior and adaptation to precision feeding technologies is needed to better utilize these technologies and further understand the relationship between feeding behavior and nutrient utilization of ruminant animals.

## Figures and Tables

**Figure 1 animals-13-02872-f001:**
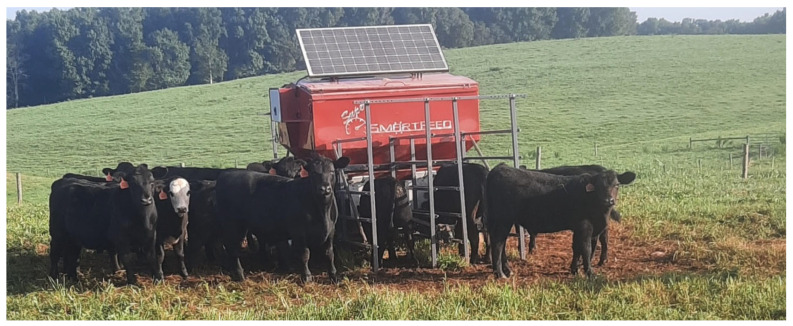
Steers training to C-Lock SuperSmart Feeder. Image Credit: J. Luke Jacobs, Clemson University.

**Figure 2 animals-13-02872-f002:**
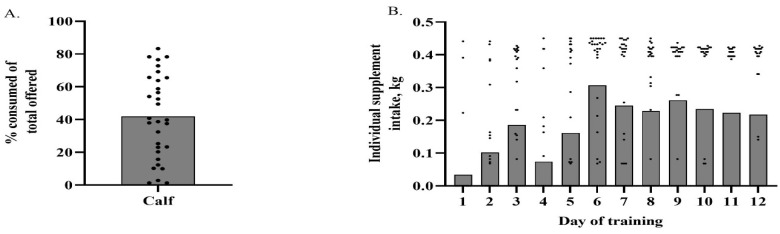
(**A**) Total percent of offered supplement consumed (kg) by individual suckling calves throughout training period relative to the group average of 41.84% (depicted by the bar). Individual dots above the line represent calves that successfully trained to the Super SmartFeeder. (**B**) Daily individual calf intake over the course of the training period. Individual calves are depicted as dots and the group average by day of training is depicted as a bar.

**Figure 3 animals-13-02872-f003:**
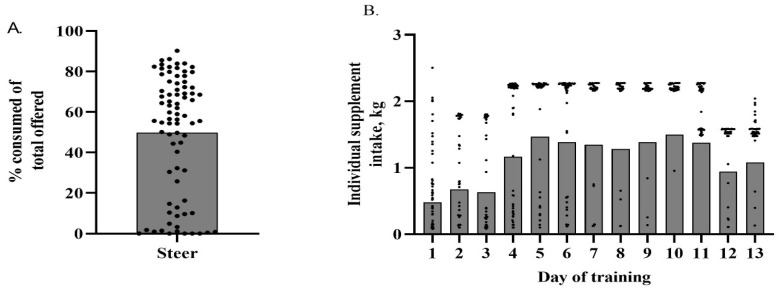
(**A**) Total percent of offered supplement consumed (kg) by individual weaned steer throughout training period relative to the group average of 49.78% (depicted by the bar). Individual dots above the line represent steers that successfully trained to the Super SmartFeeder. (**B**) Daily individual steer intake over the course of the training period. Individual steers are depicted as dots and the group average by day of training is depicted as a bar.

**Figure 4 animals-13-02872-f004:**
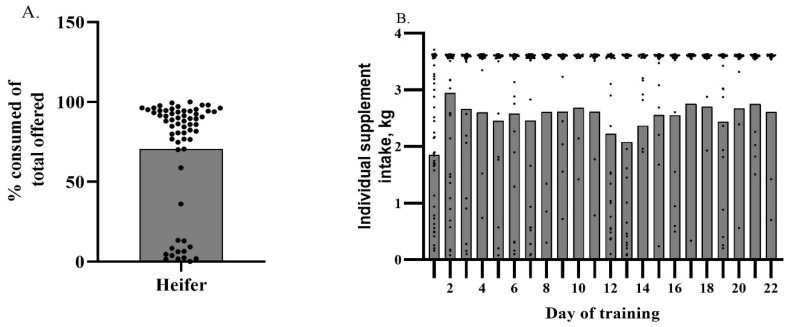
(**A**) Total percent of offered supplement consumed (kg) by individual replacement heifer throughout training period relative to the group average of 70.64% (depicted by the bar). Individual dots above the line represent heifers that successfully trained to the Super SmartFeeder. (**B**) Daily individual heifer intake over the course of the training period. Individual heifers are depicted as dots and the group average by day of training is depicted as a bar.

## Data Availability

Not applicable.

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
