# Peer review of "Training and Adaptation of Beef Calves to Precision Supplementation Technology for Individual Supplementation in Grazing Systems"

_animals, 2023, doi:10.3390/ani13182872_

Round 1

Reviewer 1 Report

This manuscript addresses the time it takes to adapt cattle to supplementation using precision supplementation equipment (super smartfeed). Very little literature is available that directly quantifies the time till adaptation, therefore, this work provides significant scientific merit and warrants publication. In general, comments are related to formatting. Specifically, Figures 2 - 4 are not located in the appropriate locations and are very small and difficult to read. Please consider providing larger figures. Although there is limited literature available related specifically to training animals to precision systems, the literature cited in this manuscript seems minimal. There are multiple publications utilizing smartfeed systems that have quantified variation of supplement intake for different age classes of beef cattle that may be useful for some of the introduction and discussion. Additionally, the lack of quantifying training in these studies would add further support to this specific manuscript. Please see line by line comments below:

160 – 175 Figure 2 is located in the middle of a paragraph. Please move to the end of the paragraph. Figure 3 should be moved to the end of the following paragraph.

185 – 192 Paragraph structure needs to be corrected, hard returns in middle of sentences.

205 revise sentence “where and decreased until d 13”

Figure 4 needs to be moved to the end of the paragraph it was first referenced.

208 This section seems out of place, I would recommend creating a discussion section here, or moving the paragraph to the end of the suckling calf results.

271 – 273 Consider replacing “younger” and “older” with the categories of cattle used in this study, as cattle at varying ages not included in this study may not be as successful as yearling heifers.

Author Response

Reviewer 1

This manuscript addresses the time it takes to adapt cattle to supplementation using precision supplementation equipment (super smartfeed). Very little literature is available that directly quantifies the time till adaptation, therefore, this work provides significant scientific merit and warrants publication. In general, comments are related to formatting. Specifically, Figures 2 - 4 are not located in the appropriate locations and are very small and difficult to read. Please consider providing larger figures. Although there is limited literature available related specifically to training animals to precision systems, the literature cited in this manuscript seems minimal. There are multiple publications utilizing smartfeed systems that have quantified variation of supplement intake for different age classes of beef cattle that may be useful for some of the introduction and discussion. Additionally, the lack of quantifying training in these studies would add further support to this specific manuscript. Please see line by line comments below:

Thank you.

160 – 175 Figure 2 is located in the middle of a paragraph. Please move to the end of the paragraph. Figure 3 should be moved to the end of the following paragraph.

Corrected.

185 – 192 Paragraph structure needs to be corrected, hard returns in middle of sentences.

Corrected.

205 revise sentence “where and decreased until d 13”

Removed “where” to clarify sentence.

Figure 4 needs to be moved to the end of the paragraph it was first referenced.

Corrected.

208 This section seems out of place, I would recommend creating a discussion section here, or moving the paragraph to the end of the suckling calf results.

Moved to end of suckling calf results as you suggested.

271 – 273 Consider replacing “younger” and “older” with the categories of cattle used in this study, as cattle at varying ages not included in this study may not be as successful as yearling heifers.

Addressed by specifying age class of cattle being referenced.

Reviewer 2 Report

As indicated in the manuscript, diet may have been a major factor when adapting the cattle to the trailers. However, this was not addressed in the discussion. Lower quality diets tend to aid in adaptation, please discuss. 

In many cases, the papers cited were using animals in the western US, where diets are low in CP, which greatly impacts intake of supplement. South Carolina is not a great representation for the typical supplementation environment. Limitations of this should be at least be mentioned.

There was a change in time to lowering the gates between the steers and heifers, why? Also, on the suckling study, allowing cows access to the trailer may have aided in training the calves because calves rely on the cows to teach them where and what to eat.

Why was the feed changed in the suckling calf study on d 4?

P-values 2 or 3 decimal places is sufficient.

Check spacing and paragraphs for the final document, there were some odd paragraph breaks after the figures.

What was your non-feeder rate? Were those accounted for?

Less than 90d of age is pretty young to begin supplementation, older calves may have worked better.

This was a single year study, this should be repeated because environment has a large impact on supplementation behavior.

I think the minimum number of references is 30 for the journal, several more need to be added. I would suggest looking for Samuel Wyffels as an option, he has conducted a lot of work with the C-Lock trailers.

Author Response

Reviewer 2

As indicated in the manuscript, diet may have been a major factor when adapting the cattle to the trailers. However, this was not addressed in the discussion. Lower quality diets tend to aid in adaptation, please discuss. 

Cattle used in the three experiments were managed according to Clemson Beef Farm SOPs, and supplemention from the SuperSmart Feeder was added to “normal” operations. The goal of this series of experiments was not to evaluate diet quality differences, but to look at adaptation of cattle from various production stages to a supplement delivery system.

In many cases, the papers cited were using animals in the western US, where diets are low in CP, which greatly impacts intake of supplement. South Carolina is not a great representation for the typical supplementation environment. Limitations of this should be at least be mentioned.

The goal of this series of experiments was to evaluate the adaptation of cattle from various production stages to a supplement delivery system. However, supplementation of cattle in the south-eastern U. S. is relatively common due to producer accessibility to cattle. Many producers offer creep feed to suckling calves, provide supplement to calves during weaning, and supplement replacement heifers to ensure nutrient requirements are met and to meet desired production goals. The weaned steers were placed in a dry lot with ad libitum bermudagrass hay with quality reported, as stated in the materials and methods section 2.2.2., to replicate common weaning practices of South-eastern producers and Clemson Beef Farm SOP. Both the suckling calves and replacement heifer experiments were conducted on tall fescue pastures, which replicates production systems faced by producers throughout much of the cow-calf oriented states in the U. S., as there are approximately 35 million acres of tall fescue, covering 15 states, and is home to approximately 40% of the U. S. cow-calf operations as reported by the USDA in 2019. These suggest that this series of experiments is representative of a large portion of the U.S., even though most of the cited studies occurred in the western U. S. The main reason for the location of most of the cited work is the relatively limited number of these SuperSmart Feeder units in the Southeast.

U.S. Department of Agriculture. Quick Stats. Washington, DC: U.S. Department of Agriculture, National Agricultural Statistics Service, 2019. Available online at https://quickstats.nass. usda.gov/

There was a change in time to lowering the gates between the steers and heifers, why? Also, on the suckling study, allowing cows access to the trailer may have aided in training the calves because calves rely on the cows to teach them where and what to eat.

Each experiment was run independently, and all had unique challenges that factored into the training method. With the suckling calf experiment, the only way we could ensure individual supplementation by the calf was to exclude the cow from being able to access the feeder by leaving the gates down. It is a relatively common practice to creep feed calves in which the cow is excluded from the creep feeder by gates and the calves figure out on their own how to eat. The steer study involved using weaned calves who did not have previous exposure to creep or supplementation. The heifers were previously exposed to supplementation and were used to eating out of bunks during the backgrounding period.

Why was the feed changed in the suckling calf study on d 4?

The initial commodity-based supplement was offered as a more palatable feed to encourage consumption and feeder utilization. A statement was added to the Materials and Methods section to clarify.

P-values 2 or 3 decimal places is sufficient.

P-values reduced to three decimal places.

Check spacing and paragraphs for the final document, there were some odd paragraph breaks after the figures.

Corrected.

What was your non-feeder rate? Were those accounted for?

Yes, those cattle were accounted for. Non-feeder cattle of all three studies were included in all statistical analyses, as stated in the Statistical Analysis section on lines 145 and 146, and were classified in the non-adopter groups. These cattle were also represented in Figures 2A, 3A, and 4A by dots along the x-axis.

Less than 90d of age is pretty young to begin supplementation, older calves may have worked better.

The suckling calf experiment was conducted to evaluate if the SuperSmart Feeder could be used as a creep feeder and gain individual intake data on suckling calves. It is always a challenge with creep feeding experiments to figure out which calves actually consumed the creep feed. No information was available on using the SSF in suckling calves so we examined this. This would be the typical time period when creep feed would be offered to suckling calves in our production system.

This was a single year study, this should be repeated because environment has a large impact on supplementation behavior.

These were independent studies conducted to determine how cattle trained to the SSF over a three-year period. The first issue with designing experiments using the SSF was deciding how best to train them to use the feeder. No information existed on the best way to train so we thought it was valuable to evaluate the training period and publish this information. The use of the SSF allows us to monitor intake of supplements in forage grazing systems, which is one of the most important issues when trying to evaluate supplementation programs. We submitted this as a communication instead of an article to provide important information to others on what we learned about training animals of various ages and management systems on the SSF.

I think the minimum number of references is 30 for the journal, several more need to be added. I would suggest looking for Samuel Wyffels as an option, he has conducted a lot of work with the C-Lock trailers.

The authors submitted the work as a communication, instead of an article, due to its relatively short nature and it being a composition of findings from a collection of studies. No indication of a minimum number of citations were observed in the author instruction packet for communications.

Reviewer 3 Report

This is awesome data.  We need this type of research as we move further into the precision ag and individual animal management era.

I have a couple of nits to pick.  Upon first reading, I did not understand why success was defined as total individual intake greater than or equal to the average percentage intake of total supplements offered to the respective group (Line 109-110).  After I read your citations, specifically Husz et al. (2020), it made perfect sense.  I think that adding a paraphrased version of this, “If the animal was eligible for feed, the supplement was dispensed in 30-s intervals until the imposed 0.50 kg/d supplement allotment was dispensed,” will improve understanding of the rationale behind using feed consumed per day as the metric for feeding success.  I pulled that sentence from page 869 of Husz et al. (2020).

Part of where I got hung up is I am familiar with the GreenFeed system where a number of drops per day are preprogrammed into the system and cattle need to keep their head in the box for three minutes to have a recordable visit.  So, I might be the only reader who gets tripped up there.  No change is a perfectly acceptable answer to this one. 

I also think that is important for readers of your paper to understand that the feeder weighs the pan after a calf backs out.  I originally had a comment in my review about times that the feed dispensed was not consumed by the calf that originally triggered the feed drop.  You might consider expanding on quantifying feed delivered in your methods section.  

Second nit.  It has been a minute since my last foray into non-parametric statistics.  On your Chi-squared test, were you testing independence?  I am having trouble interpreting the P-value given on lines 152, 179, and 193-194 with the current information given.   For example on line 152, are you saying, "14 adapted out of 31 total is not different (P = 0.5900) than 17 non-adapted out of 31 total?"  I wonder if having these proportions in a table format would make it easier for readers to understand rather than figures 2-4 A.   

The P-values of intake observed between adapted and non-adapted make sense (lines 157, 184, and 198) and are appropriate for the analysis described in lines 145-147.   

Author Response

Reviewer 3

This is awesome data.  We need this type of research as we move further into the precision ag and individual animal management era.

Thank you.

I have a couple of nits to pick.  Upon first reading, I did not understand why success was defined as total individual intake greater than or equal to the average percentage intake of total supplements offered to the respective group (Line 109-110).  After I read your citations, specifically Husz et al. (2020), it made perfect sense.  I think that adding a paraphrased version of this, “If the animal was eligible for feed, the supplement was dispensed in 30-s intervals until the imposed 0.50 kg/d supplement allotment was dispensed,” will improve understanding of the rationale behind using feed consumed per day as the metric for feeding success.  I pulled that sentence from page 869 of Husz et al. (2020).

The definition of success in a study like this can be set at many different levels (easy, medium or hard). For our purpose, we were interested in assessing how the cattle of each group adapted to the use of the SSF. Therefore, we used a medium level of stringency for determining adaption. It was not that they just accessed the feeder (easy) but that they accessed the feeder and consumed a threshold level of feed each day (above the average of the group). We felt that this reflected our goal to examine training and adaption of cattle to the SSF. For the dispensing of feed by SSF, a comment was added for clarity.

Part of where I got hung up is I am familiar with the GreenFeed system where a number of drops per day are preprogrammed into the system and cattle need to keep their head in the box for three minutes to have a recordable visit.  So, I might be the only reader who gets tripped up there.  No change is a perfectly acceptable answer to this one. 

Unlike the GreenFeed system, the SSF allows us to program drops of feed as the animal accesses the feeder. For the SSF, it records the weight of each drop as the animal accesses the feeder for that visit. It will continue to drop feed as the animal is at the feeder as recorded by the RFID tag until they reach their maximum allotment for the day. An animal may make several trips to the feeder to consume their supplement or eat it all at one time. We are currently investigating the behavior of the animals and how the animal consumed the supplement to learn more about the use of the SSF but this is for a supplementation study and not for the training period as examined in this study.

I also think that is important for readers of your paper to understand that the feeder weighs the pan after a calf backs out.  I originally had a comment in my review about times that the feed dispensed was not consumed by the calf that originally triggered the feed drop.  You might consider expanding on quantifying feed delivered in your methods section.  

The feeder was set-up to deliver a small amount of feed at a time (< 0.1 kg per drop) when they accessed the feeder. This method of supplement delivery leaves very little residual feed in the pan and helps ensure that they consume all the supplement dropped.

Second nit.  It has been a minute since my last foray into non-parametric statistics.  On your Chi-squared test, were you testing independence?  I am having trouble interpreting the P-value given on lines 152, 179, and 193-194 with the current information given.   For example on line 152, are you saying, "14 adapted out of 31 total is not different (P = 0.5900) than 17 non-adapted out of 31 total?"  I wonder if having these proportions in a table format would make it easier for readers to understand rather than figures 2-4 A.   

Correct, the chi-square test was used to test the independence of the two categories in which the cattle were assigned.

The P-values of intake observed between adapted and non-adapted make sense (lines 157, 184, and 198) and are appropriate for the analysis described in lines 145-147.   

Thank you.

Reviewer 4 Report

General

comments

Although automated individual feeding units provide a useful tool for precision supplementation of individual animal in grazing systems, knowledge is needed in how to best use these tools.  As a result, the information in this paper makes a good contribution to the literature.  The paper is well-written and the design and analysis within the system evaluated in each experiment is appropriate.  However, the variables of pasture size supplement composition, supplementation experience, and length of the training period varied with each class of cattle possibly as a result of the system for managing a specific class of cattle.  These variables confound the results between experiments making any comparisons between the different age groups questionable unless it can be established how the specific management practices were intrinsic to each production stage.  One item that should be clarified in the methods section is the previous experience that the heifers in experiment 3 had with supplementation. This is mentioned in line 258 and 259 of the results, but isn’t discussed in the methods.  Other comments follow:

Line

11

Change to ‘affecting’

40-41

While this conclusion inherently makes sense, it seems inappropriate here because of the confounding of variables in the experiment.

114-139

The differences in training period, pasture size, supplement composition, and additional supplementation outside of the SSF stalls confound results between experiments.  Where possible, the authors might explain the reasons for management variables within each experiment.

136

In lines 258 and 259, it is implied that the heifers had supplementation experience, but it is not mentioned in the methods.

161, 185, 188

These sentences are broken by spaces.

251-252

Use of heifers that will not use an automatic feeder as a control animals in study would seem to be at best questionable.  It certainly would not be a random sample and may be the result of behavioral issues that might affect growth and feed efficiency.

258

There was no discussion of previous exposure to supplementation in the methods section.

269

Because the length of training, pasture size, supplement type and experience were confounded with production stage, this conclusion seems inappropriate for this experiment.  It might be help if the authors explained the necessity of how the differences in those variables for the different production stages.

Author Response

 Reviewer 4

Suggestions for Authors

General

comments

Although automated individual feeding units provide a useful tool for precision supplementation of individual animal in grazing systems, knowledge is needed in how to best use these tools.  As a result, the information in this paper makes a good contribution to the literature.  The paper is well-written and the design and analysis within the system evaluated in each experiment is appropriate.  However, the variables of pasture size supplement composition, supplementation experience, and length of the training period varied with each class of cattle possibly as a result of the system for managing a specific class of cattle.  These variables confound the results between experiments making any comparisons between the different age groups questionable unless it can be established how the specific management practices were intrinsic to each production stage.  One item that should be clarified in the methods section is the previous experience that the heifers in experiment 3 had with supplementation. This is mentioned in line 258 and 259 of the results, but isn’t discussed in the methods.  Other comments follow:

Information on previous supplementation for each group of cattle was added to the methods.

Line

11

Change to ‘affecting’

Corrected

40-41

While this conclusion inherently makes sense, it seems inappropriate here because of the confounding of variables in the experiment.

Clarified the conclusion.

114-139

The differences in training period, pasture size, supplement composition, and additional supplementation outside of the SSF stalls confound results between experiments.  Where possible, the authors might explain the reasons for management variables within each experiment.

Each study was run independently and cattle that had used the SSF were never used again for a different study. Each study had its own unique challenges due to the location of forage system available for grazing at that the time of the study and the need to exclude other animals as in the calf experiment. The SSF needs to access wi-fi to report data to the cloud and therefore it had to be placed in paddocks where this was possible for our farm unit. We followed our normal farm management for these experiments and added the SSF to examine how cattle adapted to it. One of the issues with the SSF, is that the cattle have to use it in order to get information on individual intake and no one knew how many cattle you would need to start with in order to get a specific number for a subsequent experiment. We thought it was important to communicate our results and observations to others so that they had a starting point to work from when using the SSF for experiments.

136

In lines 258 and 259, it is implied that the heifers had supplementation experience, but it is not mentioned in the methods.

Added the previous heifer supplementation experience to materials and methods section 2.2.3..

161, 185, 188

These sentences are broken by spaces.

Corrected.

251-252

Use of heifers that will not use an automatic feeder as a control animals in study would seem to be at best questionable.  It certainly would not be a random sample and may be the result of behavioral issues that might affect growth and feed efficiency.

This series of experiments focused on observing cattle adaptation to supplement delivery systems, and therefore no treatments were assigned to animals in this series of experiments. Animals were classified as successfully trained or failed to train based on their individual usage of the SuperSmart Feeder.

258

There was no discussion of previous exposure to supplementation in the methods section.

Added prior supplementation exposure to materials and methods.

269

Because the length of training, pasture size, supplement type and experience were confounded with production stage, this conclusion seems inappropriate for this experiment.  It might be help if the authors explained the necessity of how the differences in those variables for the different production stages.

Each study was run independently and had its own unique challenges due to the location of forage system available for grazing at that the time of the study and the need to exclude other animals as in the calf experiment. We followed our normal farm management plan for these experiments and just added the SSF to examine how cattle adapted to it. Many factors can influence supplementation systems in cattle production but one of the great challenges is the ability to get individual supplementation values. The SSF provides us the opportunities to collect this information but also presents challenges on how best to train animals to use it. Our focus was to examine how the animals would adapt to the use of the SSF in normal production systems and report our observations to others so that they had a starting point to work from when using the SSF for experiments.